# Effects of Gestational Diabetes in Cognitive Behavior, Oxidative Stress and Metabolism on the Second-Generation Off-Spring of Rats

**DOI:** 10.3390/nu13051575

**Published:** 2021-05-08

**Authors:** Maribel Huerta-Cervantes, Donovan J. Peña-Montes, Miguel Ángel López-Vázquez, Rocío Montoya-Pérez, Christian Cortés-Rojo, María Esther Olvera-Cortés, Alfredo Saavedra-Molina

**Affiliations:** 1Instituto de Investigaciones Químico-Biológicas, Universidad Michoacana de San Nicolás de Hidalgo, Morelia 58030, Michoacán, Mexico; marzy112@yahoo.com.mx (M.H.-C.); yodonnie@gmail.com (D.J.P.-M.); rmontoya@umich.mx (R.M.-P.); christian.cortes@umich.mx (C.C.-R.); 2Centro de Investigación Biomédica de Michoacán, Instituto Mexicano del Seguro Social, Morelia 58341, Michoacán, Mexico; migangelv@yahoo.com.mx

**Keywords:** gestational diabetes, second-generation, offspring, anxiety, hippocampus, cerebral cortex, spatial working memory, metabolism, oxidative stress

## Abstract

Gestational diabetes (GD) has a negative impact on neurodevelopment, resulting in cognitive and neurological deficiencies. Oxidative stress (OS) has been reported in the brain of the first-generation offspring of GD rats. OS has been strongly associated with neurodegenerative diseases. In this work, we determined the effect of GD on the cognitive behavior, oxidative stress and metabolism of second-generation offspring. GD was induced with streptozotocin (STZ) in pregnant rats to obtain first-generation offspring (F1), next female F1 rats were mated with control males to obtain second-generation offspring (F2). Two and six-month-old F2 males and females were employed. Anxious-type behavior, spatial learning and spatial working memory were evaluated. In cerebral cortex and hippocampus, the oxidative stress and serum biochemical parameters were measured. Male F2 GD offspring presented the highest level of anxiety-type behavior, whilst females had the lowest level of anxiety-type behavior at juvenile age. In short-term memory, adult females presented deficiencies. The offspring F2 GD females presented modifications in oxidative stress biomarkers in the cerebral cortex as lipid-peroxidation, oxidized glutathione and catalase activity. We also observed metabolic disturbances, particularly in the lipid and insulin levels of male and female F2 GD offspring. Our results suggest a transgenerational effect of GD on metabolism, anxiety-like behavior, and spatial working memory.

## 1. Introduction

It is estimated that 21.3 million live births worldwide are affected by some type of hyperglycemia in pregnancy, of which 83% are due to GD and currently, one in six pregnancies are affected by GD [1]. The transport of glucose across the placenta affects the pregnant woman’s metabolic status and interferes with fetal development [2]. Diabetes during pregnancy causes an abnormal intrauterine metabolic state that in extreme cases results in congenital malformations and neonatal hypoglycemia depending on the severity of diabetes [3,4]. These alterations have short- and long-term consequences, including birth weight changes, insulin resistance, obesity, and diabetes [5,6]. Exposure to an adverse intrauterine environment, including hyperglycemia and hyperinsulinemia, programs the development of offspring in a way that can affect more than one generation through epigenetic changes [7]. Previous studies have shown that second-generation offspring (F2) exposed to malnutrition (low-protein or high-fat diet) or obesity during the prenatal stage present alterations in carbohydrate metabolism [8,9,10]. Studies carried out by Aerts et al. [11] revealed that GD overstimulates and disorganizes fetal pancreatic β-cell function in the offspring; consequently, there are alterations in glucose metabolism at different life stages and across generations (transgenerational effect).

In addition to metabolic alterations, it has been reported that GD modifies normal brain development; children of mothers with diabetes present neurodevelopmental problems including alterations in learning ability, attention, and motor function [12,13,14,15] similar to that observed in animal models [16,17,18]. Therefore, it is possible that GD has a transgenerational effect on offspring neurodevelopment.

On the other hand, it has been widely recognized that oxidative stress in brain areas such as the hippocampus can negatively impact the normal function of the central nervous system, and it has also been considered that it can play a crucial role in the development of neurodegenerative diseases [19,20,21] and neuropsychiatric disorders such as depression or anxiety [22,23]. 

Ours laboratories recently reported the impact of GD on metabolism, cognitive ability and oxidative stress in areas of the brain associated with cognition in the first-generation offspring (F1). We observed alterations in glucose and lipid metabolism, deficiencies in spatial learning and short-term memory, modifications in anxiety-like behavior and a rise in oxidative stress resulting in oxidative injury as lipid-peroxidation; these effects are dependent on age and sex [24]. However, there is limited information about the effect of GD on F2 offspring cognitive ability. Therefore, the objective of this work was to determine if GD modifies the metabolism, anxiety-like behavior, learning and memory of its offspring of juvenile and adult second-generation males and females and the effect on oxidative stress in brain structures related to cognition such as the cerebral cortex and hippocampus.

## 2. Materials and Methods

### 2.1. Animal Care and Use Statement

All procedures used in the present study were carried out according to the Guide for the Care and Use of Laboratory Animals (NIH Publication No. 80–23) and the Official Mexican Standard for the use of experimental animals (NOM-062-ZOO-1999) and approval by the Research and Ethics Committee of the Mexican Institute of Social Security (IMSS) (R-2017–1603-16). 

### 2.2. Experimental Design and Animal Breeding

Female Wistar rats (F0) weighing 280 to 300 g were used, which were kept under a normal light–dark cycle (12/12 h) at a temperature of 22 °C in laboratory animal facility with standard rodent chow (Rat Diet 5012, Lab Diet, St Louis, MO, USA) and water ad libitum. Female rats were mated with control males; copulation was confirmed by detection of sperm in vaginal smear and was designated as gestational day 0. The pregnant females were randomly divided into two groups: control pregnant rats (GC, *n* = 6) and rats with gestational diabetes (GD, *n* = 6). GD were each induced by a single dose of STZ. The model of GD and obtaining first-generation offspring (F1) has already been previously reported [24].

Females F1 offspring (GC and GD group, *n* = 6) were mated with control males to obtain F2 offspring. Once pregnant, they were allowed to reach term and give birth spontaneously. At birth, litters were adjusted to 12 pups, and the same number of males and females were kept within each litter. After weaning, F2 offspring were kept in groups of four rats per cage with soft bedding under standard conditions of light–dark cycle, with standard rodent chow and water ad libitum. 

Second-generation offspring of GC and GD, male and female, juvenile (two months of age) and adult (six months of age) were evaluated. Behavioral tests were performed on 10 to 12 rats per group. Once the behavioral evaluation was finished, the rats (*n* = 5 to 6) were sacrificed by decapitation to biochemical determinations.

### 2.3. Cognitive Assessment

Behavioral test was performed between 8:00 and 15:00 h, following the sequential order: elevated plus maze, open field, Morris water maze, and eight arms radial maze.

#### 2.3.1. Anxious-Type Behavior Tests

##### Elevated Plus Maze Test

The maze consisted of two opposite open arms (50 × 10 cm) and two opposite closed arms (50 × 10 × 40 cm) with an open central area illuminated with 500 lux approximately. The maze was raised to 50 cm height. On the day of evaluation, the rat was placed in the center and its behavior was recorded for 10 min. The videos were analyzed with Sci-Works software (DataWave Technologies Inc., Longmont, CO, USA). The time that the rats spent in the closed arms was determined.

##### Open Field Test

The paradigm consisted of a square arena (60 × 60 cm) divided into eight quadrants (a peripheral area that included the corners), and a central area. The center of the arena was illuminated with 500 lux approximately. During the experiment, the rat was placed in the center and the behavior was recorded for 10 min. The videos were analyzed using Sci-Works software (DataWave Technologies Inc., Longmont, CO, USA) and the time that the rats spent in the central zone was measured.

#### 2.3.2. Learning and Memory Tests

##### Spatial Learning Test

Spatial learning was assessed using the Morris water maze. The maze consisted of a circular tub of 100 cm in diameter and 28 cm in height for juvenile rats and 150 cm in diameter and 50 cm in height for adults, filled with water at a temperature of 27 ± 2 °C. The water was stained with gentian violet (dark blue color). Inside the tub, a circular platform 2 cm below the water level was placed in a fixed position (target quadrant) in one of the four quadrants throughout the study. Visual cues were posted in the registration room. At the periphery of the tub, 10 starting positions were placed. The training phase consisted of two trials per day with a 20 min inter-trial period for eight consecutive days. On the day of the experiment, the rat was placed with its face towards the wall of the tub in one of the starting positions (chosen randomly); it was released and was allowed to find the platform in a maximum time of 60 s; if it did not reach the platform, the experimenter gently guided the rat to it. Once on the platform, rats had to stay on it for 15 s. On the ninth test day, a probe trial was performed without the platform into the tub to assess memory; in this trial, the rat was released from one of the starting positions and allowed to swim for 30 s. The swim paths were analyzed using Sci-Works software (DataWave Technologies Inc., Longmont, CO, USA). Escape latency, the distance traveled during the eight test days and the time spent in each quadrant during the probe trial were measured.

##### Spatial Working Memory Test

To assess spatial working memory (short-term memory), we used the eight-armed radial maze. The maze consisted of a circular central zone (30 cm in diameter), from which eight arms extended (60 cm × 10 cm). For this behavioral test, the rats were kept under food restriction (they ate 80% of their normal intake ad libitum) for two weeks before testing. The test consisted of a two-day habituation period. On day 1 of habituation, the rats remained for 10 min in the maze with free access to exploration; on day 2, the rats were placed in the maze for 10 min with a reward (fruit cereal) available in the central area and the arms of the maze. On the third day, the training period began; the rats were placed in the maze center with a reward in the end of arms. The rats performed two trials per day with a 20-min inter-trial interval for eight consecutive days. The test ended when the rat ate the reward in all eight arms with a maximum of 10 min. The number of arms visited was recorded (when the rat entered with all four legs to the arm, it was considered one visit). The number of re-entry errors (entries to previously visited arms), errors of omission (entries to the arms without eating the reward), and total errors (re-entry errors plus errors of omission) were measured.

### 2.4. Tissue Preparation

After completion of behavioral tests, the rats were euthanized by decapitation, the brain was obtained, and blood samples were collected. The hippocampus and cerebral cortex were dissected bilaterally. A part of the tissues was homogenized with buffer (70 mM sucrose, 20 mM mannitol, 1 mM ethylene glycol tetraacetic acid (EGTA), 0.5% bovine serum albumin (BSA) and 10 mM 3-(N-Morpholino) propanesulfonic acid (MOPS), pH 7.4) to determine glutathione, reactive oxygen species and superoxide dismutase. To determine lipid peroxidation, the tissue was homogenized with 0.9% saline solution. Homogenates were stored at −70 °C until use. The protein content was determined by a modification of the Biuret assay and using bovine serum albumin as standard.

### 2.5. Biochemical Determinations

#### 2.5.1. Reactive Oxygen Species 

Reactive oxygen species (ROS) were determined using the probe 2′,7′-Dichlorodihydrofluorescein (H_2_DCFDA). 0.5 mg of hippocampal and cerebral cortex protein were resuspended separately in 2 mL of buffer (10 mM 4-(2-hydroxyethyl)-1-piperazineethanesulfonic acid (HEPES), 100 mM potassium chloride (KCl), 3 mM magnesium chloride (MgCl_2_) and 3 mM potassium dihydrogen phosphate (KH_2_PO_4_), pH 7.4) and incubated with 12.5 µM H_2_DCFDA for 15 min in an ice bath with constant agitation [25]. Basal fluorescence was recorded for one minute, then 5 mM/5 mM glutamate/malate was added as a substrate and fluorescence was recorded for 20 min at excitation/emission = 480/520 nm in a spectrophotometer (Shimadzu RF-5301PC, Kyoto, Japan). The results were expressed as arbitrary units/mg protein.

#### 2.5.2. Lipid-Peroxidation 

Lipid peroxidation was measured by thiobarbituric acid reactive substances (TBARS) [26]. A sample of 0.5 mg of hippocampal and cerebral cortex protein were resuspended in phosphate buffer (100 mM, pH 7.4) and mixed with a solution containing 0.375% thiobarbituric acid, 15% trichloroacetic acid and 0.25 M hydrochloric acid. Next, butylhydroxytoluene (BHT) 0.01% was added to the mixture to prevent non-specific chromophore formation, and incubated for 15 min at 90 °C. Later, samples were cooled at room temperature and centrifuged at 6720× *g* for 5 min. Absorbance at 532 nm was measured in a Perkin Elmer Lambda 18 UV VIS spectrophotometer (Perkin Elmer Inc., Shelton, CT, USA) and the TBARS concentration was calculated using the 156 mM^−1^ cm^−1^ molar extinction coefficient.

#### 2.5.3. Glutathione Measurement 

Glutathione was determined by an enzymatic recycling assay [27]. For the determination of total glutathione (GSHt), 0.5 mg of protein was resuspended in 0.1% Triton X and 0.6% sulfosalicylic acid and 100 mM potassium phosphate buffer with 5 mM ethylene diamine tetraacetic acid (EDTA) with a pH of 7.5. The mixture was sonicated in three cycles of sonication/ice for 20 s, then two freeze/thaw cycles were performed and centrifuged at 6500× *g*. The supernatant (100 µL) was placed in a mixture containing 100 mM potassium phosphate buffer with 5 mM EDTA, 100 µM DTNB and 0.1 units / mL of glutathione reductase (GR) and incubated for 30 s. The reaction started with 50 µM reduced nicotinamide adenine dinucleotide phosphate (β-NADPH) and was monitored for 5 min at 412 nm in a Perkin Elmer Lambda 18 UV VIS spectrophotometer (Perkin Elmer Inc., Shelton, CT, USA). Oxidized glutathione (GSSG) was obtained after derivatization of reduced glutathione (GSH) by incubating with 0.2% 4-vinylpyridine for 1 h at room temperature. GSH was calculated by subtracting GSSG from total glutathione. GSH/GSSG ratio was calculated by dividing the GSH by the GSSG.

#### 2.5.4. Superoxide Dismutase Activity

Superoxide dismutase (SOD) activity was measured by employing a 19160-SOD determination kit (Sigma, St. Louis, MO, USA) following the manufacturer’s instructions. The activity from the different samples was calculated using SOD from Escherichia coli and expressed as units of SOD/mg of protein.

#### 2.5.5. Catalase Activity

Catalase enzyme activity was determined using an oxygen-specific Clark-type electrode connected to a monitor (5300A Biological Oxygen Monitor, YSI, Yellows Springs, OH, USA) following the conversion of H_2_O_2_ to oxygen [28]. A total of 0.5 mg of protein from the different samples was resuspended in 0.1 M of potassium phosphate buffer with 5 mM EDTA (pH 7.6) at 25 °C and monitored for 1 min. Subsequently, 6 mM H_2_O_2_ was added and the conversion of H_2_O_2_ to oxygen was recorded for 2 min. Finally, 1.0 mM sodium azide was added to the reaction. Catalase enzyme activity was calculated using bovine catalase as a standard. The results were expressed as units of catalase/mg of protein.

### 2.6. Serum Parameters

Blood glucose levels were determined using a digital glucometer (Accuchek Performa System, Roche Diagnostics GmbH, Mannheim, Germany). Total cholesterol, high-density lipoprotein (HDL-cholesterol) and triglycerides were determined enzymatically using commercial kits (Química Clínica Aplicada, Amposta, Spain). Insulin concentration was measured by sandwich immunoassay (ELISA) with a kit (80-INSRT-E01, E10, ALPCO Diagnostic, Salem, NY, USA).

### 2.7. Statistical Analysis

The data were analyzed using the Prism program (GraphPad version 7.0, Inc., San Diego, CA, USA). All results were expressed as the mean ± standard error of the mean (SEM). Significant differences in body weight, serum determinations, anxiety-like behavior, and biochemicals biomarkers were determined using a Student’s *t*-test. A two-way analysis of variance (ANOVA) of repeated measures was used for data of spatial learning and spatial working memory, with the group as the between-subject factor and the training day as the intra-subject factor. Tukey’s post hoc test was performed if the differences were significant. For all analyzes, a value of *p* < 0.05 was considered significant. Finally, an analysis of the effect size (d) was conducted if the outcome was statistically significant.

## 3. Results

### 3.1. Effect of GD on the Body Weight of Second-Generation Offspring

The body weight of the F2 offspring was evaluated from birth to adulthood. The male F2 descendants of rats with GD did not present differences in weight at birth or weaning, but at juvenile age, they showed the highest body weight (*t* = 5.064, *p* = 0.0001, *d* = 2.26) compared with the F2 descendants of control rats. In adulthood, this modification was not observed. In the female F2 offspring of rats with GD, no differences in body weight were observed compared with the control F2 offspring at different ages (Table 1).

### 3.2. Effect of GD on the Anxious-Type Behavior of Second-Generation Offspring 

It was determined whether GD affects the anxious-type behavior of their F2 offspring. Behavior in the elevated plus maze and open field was evaluated. In the elevated plus maze, the time that rats spent in the closed arms was taken as an anxiety index. Juvenile and adult male and female F2 offspring of GD rats did not show significant differences in time spent in closed arms than control F2 offspring, as shown in Figure 1a,b. 

In the open field, the time the rats spent in the central zone was evaluated. Rats with a lower level of anxiety spent more time exploring the central area; in contrast, rats with high levels of anxiety showed less locomotion and exploration, with a preference to spend more time close to the walls of the open field. Male F2 juvenile offspring of GD rats spent significantly less time in the central zone than F2 offspring of control rats (*t* = 2.51, *p* = 0.021, *d* = 1.12), which is indicative of the highest level of anxiety (Figure 1c). On the contrary, the female F2 offspring of rats with GD showed the lowest level of anxiety and spent more time in the central zone than the female F2 offspring of control rats (*t* = 2.50, *p* = 0.022, *d* = 1.11), as can be observed in Figure 1d.

### 3.3. Effect of GD on the Spatial Learning of Second-Generation Offspring 

Spatial learning was determined through the performance in the Morris water maze and was evaluated using an eight-day learning curve. No significant differences were observed in the distance traveled, as shown in Figure 2. The male and female F2 offspring of rats with GD showed a learning curve similar to their respective control group in juvenile and adult age. They also spent more time in quadrant *p* during the test trial, indicating that they efficiently remembered in which quadrant the platform was located throughout the trial days (Figure 2c,d,g,h).

### 3.4. Effect of GD on the Spatial Working Memory of Second-Generation Offspring 

It was determined whether gestational diabetes modifies spatial working memory (short-term memory), the eight-arm radial maze was used. In juvenile age, male and female F2 offspring of rats with GD showed no spatial working memory deficiencies. No significant differences were observed in the number of total errors and the number of re-entry errors (Figure 3a–d). In adulthood, male F2 offspring of rats with GD did not show alterations in spatial working memory (Figure 3e,g). On the other hand, adult female F2 descendants of rats with GD in the general training curve did not show statistical differences. However, on days 7 and 8 of training, they showed a greater number of errors than their control group. A Student’s *t*-test was performed on the number of errors on day 7 (total errors *p* = 0.010, *t* = 2.85, *d* = 1.27; re-entry errors *p* = 0.023, *t* = 2.48, *d* = 1.11) and on day 8 (total errors *p* = 0.035, *t* = 2.26, *d* = 1.01; re-entry errors *p* = 0.036, *t* = 2.25, *d* = 1.00) showing a significant difference on both days. F2 offspring of GD rats made more total and re-entry errors than control offspring, as shown in Figure 3f,h.

### 3.5. Effecst of GD on Oxidative Stress Biomarkers of the Hippocampus and Second-Generation Cerebral Cortex 

It was determined whether GD affected oxidative stress biomarkers in the cerebral cortex and hippocampus. Reactive oxygen species production was first assessed using the fluorescent probe 2′,7′-Dichlorodihydrofluorescein. In the hippocampus and cerebral cortex of male and female F2 descendants of rats with GD, no significant differences were found in the amount of ROS. At juvenile and adult ages, they showed ROS production similar to their respective control group. Furthermore, lipid-peroxidation was evaluated by quantifying TBARS. In the second-generation descendant males of rats with GD, no differences were found in lipid-peroxidation of the cerebral cortex and hippocampus in juvenile and adult age rats (Table 2), as well as in the hippocampus of juvenile and adult females compared with their respective control group (Table 3). However, in the cerebral cortex of female F2 descendants of rats with GD at juvenile and adult age, a significant increase in lipid-peroxidation was observed (*p* = 0.0001, *t* = 9.03, *d* = 4.28; *p* = 0.012, *t* = 2.76, *d* = 1.26, respectively) in comparison with F2 control offspring females (Table 3).

### 3.6. Effect of GD on Glutathione Levels of the Hippocampus and Second-Generation Cerebral Cortex

As shown in Table 2, the cerebral cortex and hippocampus of the male F2 offspring of rats with GD did not show modifications in the levels of total, oxidized, reduced glutathione and in the GSH / GSSG ratio compared to the control group in both ages. Likewise, juvenile females’ glutathione status was not altered in either of the two brain regions evaluated. On the other hand, in the cerebral cortex of adult females, the highest level of oxidized glutathione (*p* = 0.04, *t* = 2.17, *d* = 1.05) and the lowest GSH/GSSG ratio (*p* = 0.019, *t* = 2.56, *d* = 1.25) were found, with no changes in the content of total glutathione and reduced glutathione, as shown in Table 3, indicating damage via oxidative stress.

### 3.7. Effects of GD on the SOD and Catalase Enzimatic Activity of the Hippocampus and Second-Generation Cerebral Cortex

To know the effect of GD on the antioxidant defense system of F2 offspring, SOD enzyme activity was evaluated. No significant differences were observed in SOD activity in the hippocampus and cerebral cortex of male and female F2 offspring of rats with GD in juvenile and adult age (Table 2 and Table 3). On the other hand, in the cerebral cortex and hippocampus of males and the hippocampus of juvenile and adult female F2 descendants of rats with GD, no differences were observed in the catalase activity enzyme than their respective control group. However, in the cerebral cortex of female F2 descendants of rats with GD, a decrease in the enzymatic activity of catalase was found in juvenile and as well as in the adult age compared to their control group (*p* = 0.04, *t* = 2.24, *d* = 1.16; *p* = 0.03, *t* = 2.39, *d* = 1.19, respectively) as show in Table 3.

### 3.8. Effect of GD on the Serum Parameters of Second-Generation Offspring 

The impact of GD on glucose and lipid metabolism in second-generation offspring was determined by quantifying the levels of glucose, insulin, total cholesterol, triglycerides, and HDL-cholesterol. Male F2 offspring of juvenile GD rats showed normal blood glucose level, increased insulin level (*p* = 0.046, *t* = 2.19, *d* = 1.10), total cholesterol (*p* = 0.001, *t* = 3.62, *d* = 1.62) and triglycerides (*p* = 0.022, *t* = 2.48, *d* = 1.11), as well as the lowest level of HDL-cholesterol (*p* = 0.0006, *t* = 4.24, *d* = 2.01). In adulthood they had a normal level of blood glucose and insulin, the highest level of total cholesterol (*p* = 0.001, *t* = 3.60, *d* = 1.57) and triglycerides (*p* = 0.020, *t* = 2.62, *d* = 1.31) as well as the lowest level of HDL-cholesterol (*p* = 0.003, *t* = 3.47, *d* = 1.68), as shown in Table 4. On the other hand, female F2 offspring of rats with GD in juvenile and adult age showed a normal blood glucose level, total cholesterol and HDL-cholesterol, the highest level of insulin (*p* = 0.046, *t* = 2.20, *d* = 1.10; *p* = 0.0019, *t* = 3.71, *d* = 1.76, respectively) and triglycerides (*p* = 0.0005, *t* = 4.35, *d* = 2.05; *p* = 0.0067, *t* = 3.11, *d* = 1.46, respectively) with respect to their control group, as can be seen in Table 4.

## 4. Discussion

GD affects the offspring in different critical periods of development during fetal and early postnatal life; these alterations increase the risk of suffering from metabolic syndrome, obesity, and type 2 diabetes in adulthood [25,29]. Previously in our laboratories, it was reported that GD alters metabolism, learning, and several biomarkers of oxidative stress in F1 offspring in a sex- and age-dependent manner [24]. However, there is little information about the effect of subsequent generations not exposed to the adverse intrauterine environment. In this work, we evaluated the effect of GD in males and females of F2 offspring; in particular, the cognitive ability and its possible relationship with oxidative stress in the hippocampus and the cerebral cortex. We also determined whether it alters the metabolism.

Neonatal macrosomia and microsomia have been observed as a dependent effect of maternal glucose concentration, whereas macrosomia occurs due to moderate hyperglycemia, and microsomy has been observed in severe hyperglycemia. These alterations have been corroborated in both humans and experimental animals [25,30,31,32]. We previously demonstrated that gestational diabetes affects the body weight of their first-generation (F1) offspring; males show low body weight from birth to adulthood.

Likewise, disturbances in the intrauterine environment, such as the intake of a low-protein diet or a high-fat diet during pregnancy, alters their F2 offspring’s birth weight due to alterations in insulin signaling and during adulthood; they present metabolic syndrome or insulin resistance [9,10]. 

In this work, the bodyweight of the F2 offspring was evaluated at different ages, as shown in Table 1. The F2 offspring of rats with GD did not show weight changes at birth. Conversely, the male offspring F2 of rats with GD at juvenile age showed the highest body weight (Table 1). They also presented an increase in insulin levels, total cholesterol, triglycerides, and a low level of HDL-cholesterol, without changes in basal blood glucose levels (Table 4). These results suggest that GD produces in their F2 male offspring signs of the metabolic syndrome [33], such as increased body weight, dyslipidemia, and an alteration in insulin levels that occurred during juvenile age and remained in adulthood. The female F2 offspring of rats with GD did not show body weight changes (Table 1), but showed insulin resistance (high insulin levels with normal blood glucose levels) and increased triglycerides levels at juvenile age, which remained in adulthood (Table 4). These results are in accordance with Aerts et al. [11]; they found that offspring of second-generation of rats with mild diabetes have a hyperinsulinemia in fetal age, in adulthood showing disruption in amino acids metabolism and diminution in the functionality of pancreatic β cell, leading to disturbances in carbohydrate and lipid metabolism. In previous work, we reported that GD affects F1 offspring insulin, triglycerides and HDL-cholesterol serum levels without changes in total cholesterol levels [24]. On the other hand, in F2 offspring, the total cholesterol levels are higher in both juvenile and adult males, indicating that metabolic alterations in subsequent generations could be more severe compared to F1 offspring exposed to maternal hyperglycemia. These outcomes support that fetal development in a diabetic intra-uterine milieu can induce a diabetogenic tendency in the offspring as well as a transgenerational effect in the metabolism and suggest that the male offspring of GD rats are more susceptible to maternal glucose disturbances. 

Epigenetic modifications have been linked with transgenerational effects in several intrauterine insults. Petropoulos et al. [34] reported changes in the DNA methylation of specific genes that affect signaling pathways involved in endocrine function, carbohydrate, and lipid metabolism, as well as insulin signaling in the placenta and liver of rats and humans exposed to GD, such as the PPAR γ receptor gene, whose function is to increase glucose utilization and favors the action of insulin in the liver; it also participates in the storage of lipids in adipose tissue. In addition, changes in adipokines leptin and adiponectin were observed—the highest and lowest expression, respectively. Such modifications could underly the metabolic alterations found in the present work. 

There is little information about the effect of GD on their offspring’s cognitive ability, and the current information has focused mainly on the effect in the first generation.

Nevertheless, in F2 and F3 offspring of rodents that were fed with a high-fat diet, a deficiency in associative learning has been reported. This cognitive alteration occurs in parallel with an altered amino acid profile (glutamate, aspartate, GABA and taurine) in the prefrontal cortex and the dorsal and ventral hippocampus, known to regulate these cognitive functions [35]. In the present study, spatial learning (mostly dependent on the hippocampus) and spatial working memory (short-term memory, dependent on the prefrontal cortex and hippocampus) were evaluated. 

In the male and female F2 offspring of rats with GD at juvenile and adult age, no alterations in spatial learning were found (Figure 2). Additionally, the oxidative stress biomarkers quantified in the hippocampus were similar to their respective control group (Table 2 and Table 3). Female F1 GD offspring at a young age showed a deficiency in spatial learning, possibly due to delayed neuronal maturation and oxidative stress in the hippocampus that was triggered by maternal hyperglycemia [24]. These results suggest that gestational GD does not alter the hippocampus’s integrity and, therefore, its function in a transgenerational manner.

In the F2 offspring of rats with GD, in the spatial working memory of juvenile and adult males, as well as of juvenile female, no alterations were observed; in contrast, the females in adulthood presented a deficiency in spatial working memory, which was evident on days 7 and 8 of training (Figure 3). Spatial working memory deficiency coincides with the oxidative damage in the cerebral cortex, which was evidenced by an increase in lipid-peroxidation levels. Lipid-peroxidation directly damages neuronal membranes due to it containing high levels of polyunsaturated fatty acids (PUFAs). Free radical attack of PUFAs leads to the formation of highly reactive electrophilic aldehydes, including malondialdehyde (MDA) and 4-hydroxy−2-nonenal (HNE) [36]. Peroxidation of membrane lipids leads to significant changes in cell integrity and membrane permeability, affecting several transmembrane processes, such as receptor activation, second messenger formation, alteration in cell homeostasis Ca^2+^, alterations in synaptic functionality and mitochondrial dysfunction [37]. These findings are relevant due to the lipid-peroxidation of the cerebral cortex observed at juvenile age and adulthood, which is reflected in the performance in the spatial working memory test. Additionally, in this brain region, we observed an alteration in glutathione redox status and diminished catalase activity in the female offspring of GD rats during adulthood (Table 3). 

Oxidative stress derived from mitochondrial dysfunction has been reported to play an essential role in the cognitive degeneration observed in metabolic disorders such as diabetes and neurodegenerative diseases such as Alzheimer’s or Huntington’s disease [38,39,40,41,42]. Furthermore, in cognitive deficiencies dependent on the prefrontal cortex, epigenetic modifications have been identified in the Catechol-O-transferase gene involved in processes such as attention, executive functions, and short-term learning [43].

Regarding anxiety-type behavior previously, we reported modifications in F1 offspring, male (in youth and adult age) and females (in adult age) which showed lower levels of anxiety-type behavior as a consequence of intrauterine hyperglycemia. In the F2 offspring of GD rats, significant changes in anxious-type behavior were found in juvenile age rats that did not remain in adulthood. Males showed the highest level of anxiety-type behavior and females the lowest level of anxiety-type behavior than their respective control offspring (Figure 1). It has been reported that in the offspring of rats that consumed a high-fat diet, there are epigenetic alterations in genes that are involved in the regulation of anxiety-type behavior in the amygdala, hippocampus, prefrontal cortex and hypothalamus [44]. 

In this context, our results in F2 offspring are in accordance with Warneke et al., who showed that females presented less anxiety and males greater anxiety in rats with a high energy diet (cafeteria diet); in addition, they found an increase in body weight, which suggests that obesity leads to behavioral alterations such as anxious-type behavior in adult rats, although the mechanisms involved in this modification are unknown [45]. In addition, an association has been reported between metabolic disorders such as diabetes or metabolic syndrome, with changes in anxiety behavior due to alterations in serotonin signaling [46,47,48].

Recently, it has been reported that a high-fat diet in rodents alters glucocorticoid signaling mechanisms in limbic regions of the brain [49]. Furthermore, perinatal exposure to a high-fat diet in adolescent rats leads to a decrease in anxiety-like behavior as a result of an alteration in the expression of the glucocorticoid receptor and of genes related to the inflammatory response in the hippocampus and the amygdala [50]. Therefore, it is possible that anxiety-type behavior alterations observed in the present work are closely related to the metabolic perturbances found. 

## 5. Conclusions 

In conclusion, these data indicate that GD has a transgenerational effect on the metabolism, spatial working memory and anxiety-like behavior. Nevertheless, future studies are required to evaluate the neuronal circuit and the neurotransmitters involved in anxiety-type behavior and to determine a possible mechanism underlying the transgenerational effect of gestational diabetes. 

## Figures and Tables

**Figure 1 nutrients-13-01575-f001:**
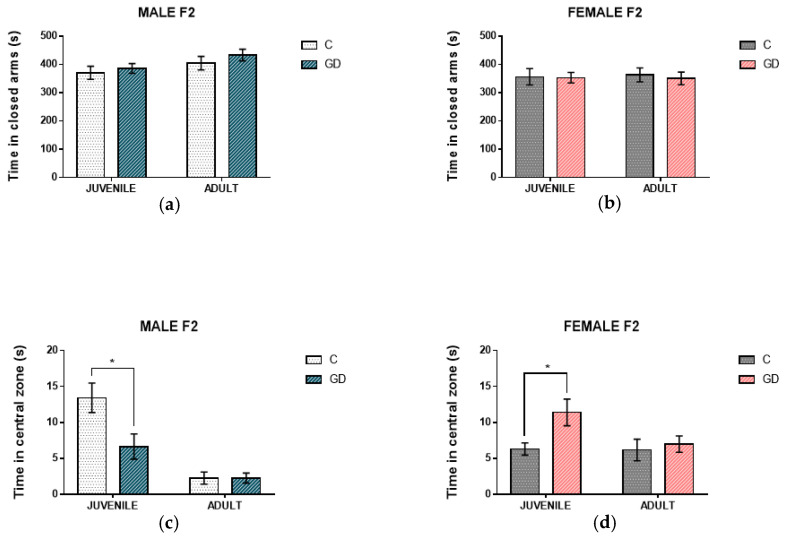
Effect of GD on the anxiety-like behavior of the second-generation offspring. The upper panel shows the time spent in the closed arms of elevated plus maze: juvenile and adult male offspring (**a**) and female offspring (**b**). The lower panel shows the time they spent in the central zone of the open field; (**c**) shows the male offspring and (**d**) the female offspring at juvenile and adult age. C, offspring of control rats; GD, offspring of rats with gestational diabetes, *n* = 10 to 12. The data are expressed as the mean ± SEM. Student’s *t*-test, * *p* < 0.05.

**Figure 2 nutrients-13-01575-f002:**
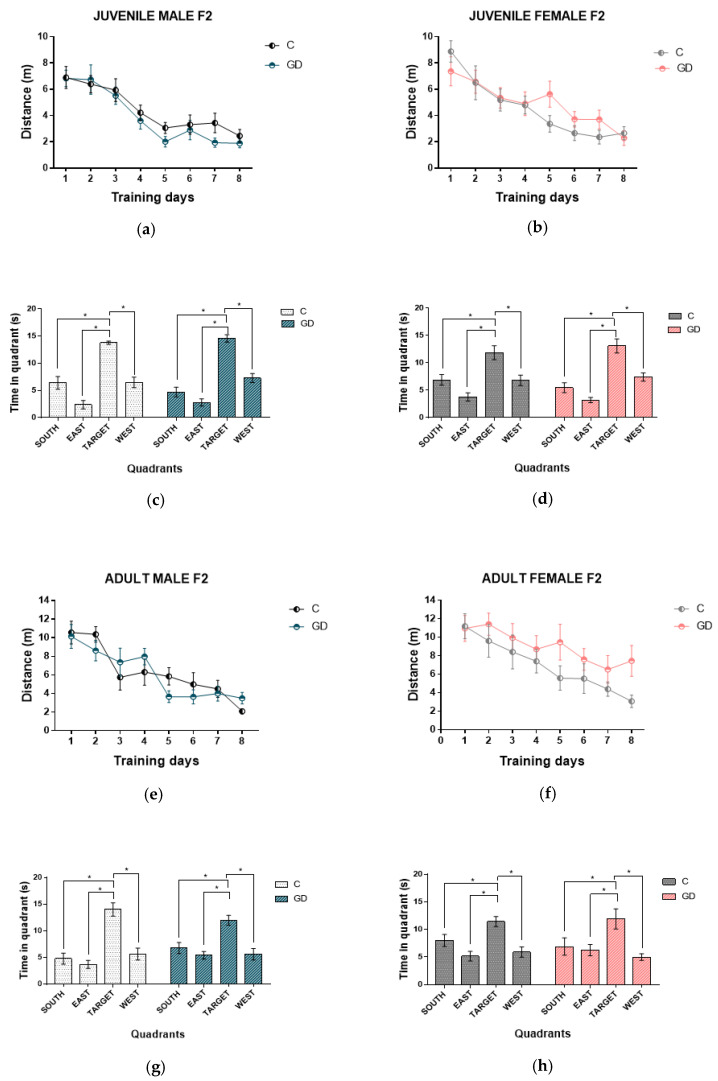
Effect of GD on spatial learning of the second-generation offspring. The distance traveled throughout the test days can be observed: male and female offspring of juvenile age (**a**,**b**) and the time they spent in each quadrant in the test (**c**,**d**). Distance traveled of the offspring in adulthood males and females (**e**,**f**) and the time in each quadrant of (**g**,**h**). C, offspring of control rats; GD, offspring of rats with gestational diabetes, *n* = 10 to 12. The data are expressed as the mean ± SEM. Two-way repeated-measures ANOVA, one-way ANOVA; Tukey’s post hoc, Student’s *t*-test, * *p* < 0.05.

**Figure 3 nutrients-13-01575-f003:**
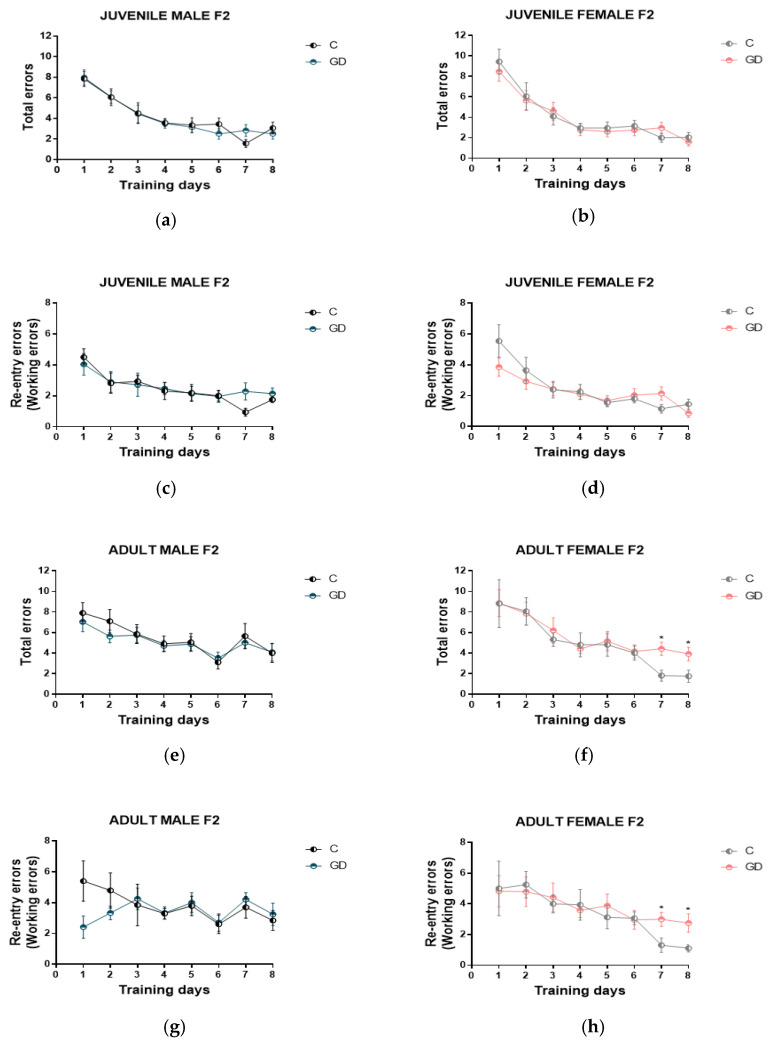
Effect of GD on the spatial working memory of second-generation offspring. The number of errors throughout the trial days can be seen. The total and re-entry errors (**a**–**d**) of the male and female offspring of juvenile and adult age (**e**–**h**) are shown. C, offspring of control rats; GD, offspring of rats with gestational diabetes, *n* = 10 to 12. The data are expressed as the mean ± SEM. Two-way repeated-measures ANOVA, one-way ANOVA; Tukey’s post hoc, Student’s *t*-test, * *p* < 0.05.

**Table 1 nutrients-13-01575-t001:** Second-generation offspring body weight at different ages.

Body Weight (g)								
Age	Male F2 Offspring		*n*		Female F2 Offspring		*n*	
	C	GD	C	GD	C	GD	C	GD
Birth	7.26 ± 0.34	7.50 ± 0.08	16	23	7.00 ± 0.31	7.53 ± 0.11	19	19
Weaning (21 days)	45.33 ± 0.79	45.63 ± 1.39	12	19	42.89 ± 0.72	46.6 ± 1.32	20	20
Juvenile (2 months)	313.9 ± 2.38	331.2± 2.44 *▪	12	10	208.4 ± 4.76	218.2 ± 3.63	12	10
Adult (6 months)	497.0 ± 8.39	500.0 ± 5.79	10	10	282.0 ± 6.40	285.8 ± 6.12	10	10

Data are expressed as the mean ± SEM. Student’s *t*-test, * *p* < 0.05, *▪ d* > 1. F2, second-generation offspring; C, offspring of control rats; GD, offspring of rats with gestational diabetes.

**Table 2 nutrients-13-01575-t002:** Oxidative stress biomarkers of male second-generation offspring.

Cerebral Cortex	Juvenile-C *n* = 6	Juvenile-GD *n* = 6	Adult-C *n* = 6	Adult-GD *n* = 6
ROS (arbitrary units/mg protein)	27.3 ± 4.71	29.0 ± 2.41	45.43 ± 2.27	51.65 ± 3.19
Lipid peroxidation (nmoles TBARS/mg protein)	263.6 ± 26.89	273 ± 28.48	203.4 ± 18.06	270.9 ± 44.75
GSHt (µmoles/mg protein)	0.31 ± 0.01	0.33 ± 0.01	0.50 ± 0.03	0.56 ± 0.04
GSSG (µmoles/mg protein)	0.12 ± 0.001	0.12 ± 0.01	0.10 ± 0.01	0.08 ± 0.008
GSH (µmoles/mg protein)	0.18 ± 0.01	0.20 ± 0.03	0.39 ± 0.03	0.47 ± 0.003
GSH/GSSG ratio	1.90 ± 0.52	1.93 ± 0.54	4.86 ± 1.14	5.74 ± 0.39
SOD activity (U/mg protein)	16.02 ± 1.83	15.0 ± 1.90	12.24 ± 2.02	12.79 ± 1.27
Catalase activity (U/mg protein)	1.39 ± 0.09	1.10 ± 0.10	1.12 ± 0.07	0.94 ± 0.09
**Hippocampus**	**Juvenile-C** ***n* = 6**	**Juvenile-GD** ***n* = 6**	**Adult-C** ***n* = 6**	**Adult-GD** ***n* = 6**
ROS (arbitrary units/mg protein)	18.01 ± 1.39	32.32 ± 8.65	41.22 ± 2.40	47.55 ± 2.49
Lipid peroxidation (nmoles TBARS/mg protein)	283.3 ± 48.42	378.9 ± 46.48	221.5 ± 27.61	195.6 ± 23.15
GSHt (µmoles/mg protein)	0.33 ± 0.03	0.30 ± 0.01	0.51 ± 0.05	0.49± 0.06
GSSG (µmoles/mg protein)	0.09 ± 0.01	0.09 ± 0.01	0.10 ± 0.01	0.11 ± 0.01
GSH (µmoles/mg protein)	0.23 ± 0.02	0.20 ± 0.01	0.40 ± 0.04	0.38 ± 0.06
GSH/GSSG ratio	2.63 ± 0.34	2.65 ± 0.47	4.34 ± 0.64	3.72 ± 0.44
SOD activity (U/mg protein)	13.19 ± 1.36	12.68 ± 2.23	13.49 ± 0.69	14.21 ± 0.52
Catalase activity (U/mg protein)	0.95 ± 0.11	0.90 ± 0.07	1.10 ± 0.05	1.13 ± 0.02

The data are expressed as the mean ± SEM. Student’s *t*-test; C, control offspring; GD, gestational diabetes offspring (*n* = 5–6). GSHt, total glutathione; GSSG, oxidized glutathione; GSH, reduced glutathione; ROS, reactive oxygen species; TBARS, substances reactive to thiobarbituric acid; SOD, superoxide dismutase; CAT, catalase.

**Table 3 nutrients-13-01575-t003:** Oxidative stress biomarkers of female second-generation offspring.

Cerebral Cortex	Juvenile-C *n* = 5	Juvenile-GD *n* = 5	Adult-C *n* = 5	Adult-C *n* = 5
ROS (arbitrary units/mg protein)	29.78 ± 5.22	28.1 ± 3.45	47.81 ± 4.59	51.66 ± 2.80
Lipid peroxidation (nmoles TBARS/mg protein)	149.9 ± 10.18	305.4 ± 13.02 *▪	158.3 ± 24.24	256.7 ± 24.14 *▪
GSHt (µmoles/mg protein)	0.31 ± 0.1	0.36 ± 0.01	0.52 ± 0.02	0.48 ± 0.03
GSSG (µmoles/mg protein)	0.11 ± 0.02	0.09 ± 0.01	0.09 ± 0.01	0.13 ± 0.01 *▪
GSH (µmoles/mg protein)	0.20 ± 0.02	0.26 ± 0.02	0.42 ± 0.01	0.34 ± 0.03
GSH/GSSG ratio	2.5 ± 0.6	4.9 ± 1.3	4.89 ± 0.74	2.76 ± 0.43 *▪
SOD activity (U/mg protein)	16.78 ± 1.91	16.05 ± 1.77	16.17 ± 1.71	14.44 ± 0.27
Catalase activity (U/mg protein)	1.10 ± 0.11	0.79 ± 0.07 *▪	1.12 ± 0.14	0.72 ± 0.08 *▪
**Hippocampus**	**Juvenile-C** ***n* = 5**	**Juvenile-GD** ***n* = 5**	**Adult-C** ***n* = 5**	**Adult-C** ***n* = 5**
ROS (arbitrary units/mg protein)	26.76 ± 2.38	25.48 ± 1.73	40.06 ± 2.97	48.07 ± 4.28
Lipid peroxidation (nmoles TBARS/mg protein)	377.2 ± 50.32	476.3 ± 44.34	208.0 ± 29.1	181.1 ± 16.79
GSHt (µmoles/mg protein)	0.33 ± 0.03	0.29 ± 0.01	0.40 ± 0.08	0.45 ± 0.01
GSSG (µmoles/mg protein)	0.10 ± 0.02	0.11± 0.01	0.11 ± 0.01	0.11 ± 0.008
GSH (µmoles/mg protein)	0.22 ± 0.04	0.17 ± 0.02	0.29 ± 0.08	0.34 ± 0.01
GSH/GSSG ratio	3.49 ± 1.18	2.02 ± 0.50	2.57 ± 0.74	3.36 ± 0.39
SOD activity (U/mg protein)	16.19 ± 2.56	17.47 ± 1.54	17.93 ± 2.60	13.74 ± 1.90
Catalase activity (U/mg protein)	0.86 ± 0.08	0.75 ± 0.07	1.10 ± 0.12	1.02 ± 0.05

The data are expressed as the mean ± SEM. Student’s *t*-test, * *p* < 0.05, ▪ *d* > 1. C, control offspring; GD, gestational diabetes offspring (*n* = 5–6). GSHt, total glutathione; GSSG, oxidized glutathione; GSH, reduced glutathione; ROS, reactive oxygen species; TBARS, substances reactive to thiobarbituric acid; SOD, superoxide dismutase; CAT, catalase.

**Table 4 nutrients-13-01575-t004:** Metabolic parameters in the serum of second-generation offspring.

Male Offspring				
Parameter	Juvenile-C *n* = 7	Juvenile-GD *n* = 7	Adult-C *n* = 10	Adult-GD *n* = 10
Glucose (mg/dL)	95.43 ± 4.26	100.9 ± 2.96	88.0 ± 1.29	89.9 ± 2.71
Insulin (ng/mL)	0.33 ± 0.015	0.39 ± 0.020 *▪	0.39 ± 0.028	0.41 ± 0.022
Total cholesterol (mg/dL)	58.36 ± 4.00	80.23 ± 4.53 *▪	62.65 ± 2.86	78.78 ± 3.48 *▪
HDL-cholesterol (mg/dL)	50.73 ± 2.73	39.15 ± 1.72 *▪	54.84 ± 1.94	43.95 ± 2.39 *▪
Triglycerides (mg/dL)	43.93 ± 2.87	59.91 ± 5.74 *▪	29.37 ± 2.18	43.80 ± 5.04 *▪
**Female Offspring**				
**Parameter**	**Juvenile-C** ***n* = 8**	**Juvenile-GD** ***n* = 8**	**Adult-C** ***n* = 8**	**Adult-GD** ***n* = 10**
Glucose (mg/dL)	89.63 ± 2.57	91.63 ± 3.57	87.25 ± 2.44	91.1 ± 1.69
Insulin (ng/mL)	0.33 ± 0.006	0.42 ± 0.046 *▪	0.35 ± 0.016	0.52 ± 0.048 *▪
Total cholesterol (mg/dL)	72.68 ± 4.21	75.46 ± 2.96	68.1 ± 2.78	65.16 ± 4.67
HDL-cholesterol (mg/dL)	51.13 ± 2.99	49.99 ± 1.60	51.58 ± 2.78	50.85 ± 2.46
Triglycerides (mg/dL)	46.32 ± 2.72	66.93 ± 3.87 *▪	36.73 ± 3.65	57.57 ± 5.69 *▪

The data are expressed as the mean ± SEM. Student’s *t*-test, * *p* <0.05, ▪ *d* > 1. C, offspring of control rats; GD, offspring of rats with gestational diabetes; High-density lipoprotein (HDL); (*n* = 5–6).

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
