# Peer review of "Effects of Gestational Diabetes in Cognitive Behavior, Oxidative Stress and Metabolism on the Second-Generation Off-Spring of Rats"

_nutrients, 2021, doi:10.3390/nu13051575_

Round 1

Reviewer 1 Report

In this paper, Huerta-Cervantes et al. aimed to determine the effect of GD on the cognitive ability, oxidative stress and metabolism of second-generation (F2) male and female offspring in juvenile and adult age. The authors conclude that there is a transgenerational effect of GD on metabolism, anxiety behavior, and working memory, leading to sex-dependent effects.

Despite the interesting topic, a series of major and minor issues have to be taken into consideration.

Major issues

1) Pathophysiology. Diabetes was induced with streptozo(to)cin (STZ), which is a toxic agent for the β-cells. Therefore, it is probable that it induces a type of diabetes that is similar to type 1 diabetes in humans (i.e., non-production of insulin). On the other hand, GD (once again, in humans) is characterized by insulin resistance. Please, discuss this issue.

Minor issues

1) Pathophysiology. Please, expand on the gender difference regarding the anxiety-like behavior (time spent in the central area of the open field - Figure 1).

2) Define all abbreviations during their first appearance in the manuscript. If applicable, every abbreviation has to be defined twice: once in the “Abstract” and once in the main manuscript. As an example, in “Abstract”, please, define STZ. In addition, some abbreviations are introduced more than once (e.g., GD) or the full term (e.g., gestational diabetes) is used after the introduction of the abbreviation.

3) Few typos occur throughout the manuscript. As examples, in the “Abstract”: “In short-term memory; only”. Please, locate and correct these typos.

The authors are kindly requested to answer all suggestions of the reviewers and the associate editor, one-by-one, using an itemized list.

Author Response

Rebuttal letter

Manuscript nutrients-1150258 of the authors: Maribel Huerta-Cervantes et al.

Note: The authors agree for the next changes added to the manuscript, in response to reviewer’s request.

Reviewer 1:

Major issues

1) Pathophysiology. Diabetes was induced with streptozo(to)cin (STZ), which is a toxic agent for the β-cells. Therefore, it is probable that it induces a type of diabetes that is similar to type 1 diabetes in humans (i.e., non-production of insulin). On the other hand, GD (once again, in humans) is characterized by insulin resistance. Please, discuss this issue.

R= The International Diabetes Federation classifies diabetes in pregnancy as the type of diabetes that is diagnosed before pregnancy or that was diagnosed during the first trimester of pregnancy. On the other hand, it defines gestational diabetes as the one first recognized during pregnancy, usually at week 24-28 (between the second and third trimesters). While insulin resistance is a parameter for determining gestational diabetes, a determining component of the disease is hyperglycemia as in other types of diabetes. While insulin resistance is a parameter for determining gestational diabetes, a determining component of the disease is hyperglycemia as in other types of diabetes. Other models that generate insulin resistance such as exposure to a diet high in fat, calories, or carbohydrates in most cases the hyperglycemia characteristic of gestational diabetes is not observed. In addition, these models are generated prior to gestation so it is considered a pregestational and non-gestational alteration.

There is currently no gestational diabetes model that reproduces what happens in humans. However, induction of gestational diabetes with streptozotocin is the most widely used model for this pathology and it is widely accepted to confirm the model with a fasting basal glucose intake. On the other hand, the glucose tolerance test that usually diagnoses gestational diabetes in humans is an invasive procedure that can lead to stress. It has been widely recognized that stress during pregnancy can cause various abnormalities in adulthood such as metabolic and cognitive, so that factor in this work is ruled out and we make sure that the results obtained are due to maternal hyperglycemia.

Another important component of pathology in humans is macromy or neonatal microsomy that can reproduce with this model, depending on the degree of maternal hyperglycemia obtained by modifying the dose of streptozotocin. The dose we used for the induction of the model was selected considering that a high glucose concentration was maintained constantly throughout gestation and that it maintained the viability of the fetuses, as well as their development.

Minor issues

  • Please, expand on the gender difference regarding the anxiety-like behavior (time spent in the central area of the open field - Figure 1).

R= Kindly, in page 6, lines 257-270, it was added an explanation as suggested by reviewer.

2) Define all abbreviations during their first appearance in the manuscript. If applicable, every abbreviation has to be defined twice: once in the “Abstract” and once in the main manuscript. As an example, in “Abstract”, please, define STZ. In addition, some abbreviations are introduced more than once (e.g., GD) or the full term (e.g., gestational diabetes) is used after the introduction of the abbreviation.

R= Kindly, as suggested by reviewer, all changes were added in the Abstract.

3) Few typos occur throughout the manuscript. As examples, in the “Abstract”: “In short-term memory; only”. Please, locate and correct these typos.

 R= Kindly, as suggested by reviewer, all changes were change in the Abstract.

Reviewer 2 Report

This study investigated the transgenerational effects of pharmacologically induced gestational diabetes (GD) in Wistar rats, describing changes in cognitive behavior, oxidative stress and metabolism in second generation (F2) offsprints in juvenile (two month old) and adult (six month old) age. Tests used for evaluation of cognitive ability in rats included the elevated plus maze, the open field, the Morris water maze and radial maze. The authors conclude the existence of different age and sex-related effects in F2 GD offsprints affecting metabolism parameters, oxidative stress biomarkers and cognitive behavior. In female rats they suggest a possible association between the production of oxidative stress in the cerebral cortex and the deficits observed in short-term memory. The study seems well-conducted with a convincing experimental design, and results are interesting in different ways. However, the discussion needs to be improved. Below I include my comments that should be reviewed by the authors.

The present study follows the research line established by the same research group in a previous paper in which the effects of DG in rats of the F1 generation (Huerta-Cervantes M. et al., 2020) were reported. Experimental designs were similar in both studies. However, it is surprising and somewhat disappointing that the results of the present study in F2 are hardly discussed with those found for F1, including changes at the level of cognitive behavior tests, metabolism and oxidative stress. I think this comparative is essential and it should be done in a meticulous way in order to improve understanding of which changes observed in F2 may already be evident in F1. It means that these changes could have a gestational origin that merit specific relevance.

On the other hand, the short discussion that takes place in relation to the changes observed in the biomarkers of oxidative stress is striking. Despite this, the changes that took place in females were directly related to short-term memory deficiency, which does not seem to occur in males. In my opinion, there is no experimental basis to support this conclusion, nor is there a bibliographic account that allows us to go beyond a simple hypothesis. Therefore, I think the authors should elaborate on this part of the discussion at a more detailed level.

Explicit references to the differences found between sex-dependent effects are also lacking. For example, the results shown in Figure 1 on the effects of GD on anxiety-like behavior are very striking. It is clearly shown that in juveniles the effects are opposite in males and females. Any discussion about it?

Abstract

This section includes a very detailed description of the observed effects, as well as the statistical results for each of the parameters studied. In my view, such a statistical description is more typical of the results section than of the abstract, and in the current way it seems exaggerated. On the other hand, authors must take into account the instructions for authors that appear in the journal. It is indicated that the maximum length of the abstract should be 200 words, which is clearly exceeded in this section.

The abstract does not include anywhere the brain tissues in which oxidative stress has been investigated, the cerebral cortex and hippocampus. Please make an explicit reference in this regard. In relation to the summary of the results, specific mentions should be made of the regions that were affected in relation to changes in the oxidative stress parameters.

Lns, 105-107. The text is not understood, perhaps a word is missing.

Ln 100-101. It is indicated that “pregnancy was verified by sperm in the vaginal smears”? Sure that is so?

Ln 423-426. The text is not well understood. It could be that the sentence is not well constructed or contains the wrong words.

Author Response

Rebuttal letter

Manuscript nutrients-1150258 of the authors: Maribel Huerta-Cervantes et al.

Note: The authors agree for the next changes added to the manuscript, in response to reviewer’s request.

Reviewer 2:

R= Kindly, all observations were made to the manuscript. In the Abstract section, changes were added as suggested and considering the maximum length of 200 words; as suggested, changes were corrected for lines 100-101; lines 105-107; lines 423-426. As well, the Discussion section was added with new information and expanded.

Round 2

Reviewer 2 Report

I have no new comments on this paper